# Taxonomy and Phylogeny of *Peniophora* Sensu Lato (Russulales, Basidiomycota)

**DOI:** 10.3390/jof9010093

**Published:** 2023-01-08

**Authors:** Yun-Lin Xu, Yan Tian, Shuang-Hui He

**Affiliations:** 1School of Ecology and Nature Conservation, Beijing Forestry University, Beijing 100083, China; 2Kenli Branch of Dongying Natural Resources and Planning Bureau, Dongying 257500, China

**Keywords:** corticioid fungi, *Dendrophora*, *Duportella*, Peniophoraceae, wood-decaying fungi

## Abstract

*Peniophora* is an old corticioid genus, from which two small satellite genera, *Dendrophora* and *Duportella*, were derived based on morphological differences. Molecular systematic studies showed that they belong to Peniophoraceae, Russulales, but the inter- and intra-generic phylogenetic relationships are still unclear. Moreover, the species diversity of this group in subtropical and tropical Asia has not been sufficiently investigated and studied. In this study, we carried out an intensive taxonomic and phylogenetic study on *Peniophora* sensu lato based on analyses of concatenated ITS1-5.8S-ITS2 (ITS, Internal Transcribed Spacer) and 28S (D1–D2 domains of nuc 28S rDNA) sequence data of all available species worldwide. In the phylogenetic trees, species of *Peniophora* s. l. (sensu lato) including types of *Peniophora* s.s. (sensu stricto), *Dendrophora* and *Duportella* were interspersed within a strongly supported clade. It means that the morphological delimitations of the three genera are not reliable, and they should be regarded as a large genus. As a result, eight species of *Duportella* were transferred to *Peniophora*, although five of them have not been sequenced. Four new distinct lineages, corresponding to *Peniophora cremicolor*, *P. major*, *P. shenghuae* and *P. vietnamensis* spp. nov., were recovered in the trees. *Peniophora taiwanensis* is treated as a later synonym of *P. malaiensis* based on morphological and molecular evidence. *Duportella renispora* is excluded from Peniophoraceae, because BLAST results of the ITS and 28S sequences of the holotype showed that it is closely related to *Amylostereum*. Descriptions and illustrations are provided for the four new species, and an identification key is given for all 25 species of *Peniophora* in China. Our results indicated that the species diversity of the corticioid fungi in Asia is rich and deserves further study.

## 1. Introduction

The genus *Peniophora* Cooke (Peniophoraceae, Russulales) introduced in 1879, typified by *Thelephora quercina* Pers. ex Fr., is one of the oldest genera of corticioid fungi. It is a cosmopolitan genus with a wide distribution from boreal to tropical areas, causing a white rot on both angiosperms and gymnosperms. Species of the genus prefer to grow on small branches especially those dead but still attached ones in the exposed and dry environments. At the beginning, mycologists adopted a broad concept, and many unrelated species were described in the genus. It includes 639 names in Index Fungorum (http://www.indexfungorum.org/Names/Names.asp, accessed on 20 November 2022), but most species have been moved to other genera, and the morphological circumscription of *Peniophora* has been narrowed [1]. Meanwhile, some intra-generic taxa were proposed and used, and two satellite genera *Duportella* Pat. and *Dendrophora* (Parmasto) Chamuris were separated from *Peniophora* s.l. and accepted by many mycologists [2,3,4,5]. In the morphological monograph of Peniophoraceae [5], Andreasen and Hallenberg accepted 70 species of *Peniophora* s.s., 12 species of *Duportella* and 2 species of *Dendrophora* and provided keys and descriptions for all the genera and species. 

According to the modern concept, *Peniophora* s.s. includes species possessing resupinate basidiomes with a smooth hymenophore, a monomitic hyphal system with simple-septate or nodose-septate generative hyphae, both encrusted cystidia and gloeocystidia, and thin-walled, smooth basidiospores negative in Melzer’s reagents. *Dendrophora* differs from *Peniophora* s.s. by having effused-reflexed basidiomes and the presence of brown, thick-walled dendrohyphidia, while *Duportella* differs by the presence of brown lamprocystidia and skeletal hyphae (in some species). However, there are usually morphological overlaps among the three genera. Based on the analyses of ITS sequences of Aphyllophorales, Boidin et al. [4] found that species of *Peniophora* s.l. formed a relatively strongly supported group that they named ‘Peniophorales’. Later, phylogenetic studies showed that *Peniophora* s.l. belongs to Peniophoraceae, Russulales [6,7,8]. Nevertheless, no comprehensive molecular studies have been carried out for *Peniophora* s.l., and the inter- and intra-generic phylogeny is not clear.

The ITS and 28S sequences of many species of *Peniophora* s.l. including some from type specimens were released in GenBank by Vu et al. [9] and thus made it possible to study the phylogeny of this group. In this paper, we studied the taxonomy and phylogeny of *Peniophora* s.l. by adding many sequence data of species from Asia, which was shown to be rich in the species diversity of this group [10,11,12,13,14,15,16]. We aim to resolve the phylogenetic relationships among *Peniophora* s.s., *Dendrophora* and *Duportella*, and explore the species diversity in China. This research is a part of the studies of species diversity, taxonomy and phylogeny of corticioid fungi in China.

## 2. Materials and Methods

### 2.1. Morphological Studies

Voucher specimens and strains are deposited at the herbaria of Beijing Forestry University, Beijing, China (BJFC) and Centre for Forest Mycology Research, U.S. Forest Service, Madison, Wisconsin, USA (CFMR). Freehand sections were made from dried basidiomes and mounted in 2% (*w*/*v*) potassium hydroxide (KOH) with 1% (*w*/*v*) phloxine, Melzer’s reagent (IKI) or cotton blue (CB). Microscopic examinations were carried out with a Nikon Eclipse 80i microscope (Nikon Corporation, Tokyo, Japan) at magnifications up to 1000×. Drawings were made with the aid of a drawing tube. The following abbreviations are used: L = mean spore length, W = mean spore width, Q = L/W ratio, *n* (a/b) = number of spores (a) measured from number of specimens (b). Color codes and terms follow Kornerup and Wanscher [17].

### 2.2. DNA Extraction and Sequencing

A CTAB plant genomic DNA extraction Kit DN14 (Aidlab Biotechnologies Co., Ltd., Beijing, China) was used to extract total genomic DNA from dried specimens which were then amplified by the polymerase chain reaction (PCR), according to the manufacturer’s instructions. The ITS1-5.8S-ITS2 region was amplified with the primer pair ITS5/ITS4 [18] using the following protocol: initial denaturation at 95 °C for 4 min, followed by 34 cycles at 94 °C for 40 s, 58 °C for 45 s and 72 °C for 1 min, and final extension at 72 °C for 10 min. The 28S D1-D2 region was amplified with the primer pair LR0R/LR7 [19] employing the following procedure: initial denaturation at 94 °C for 1 min, followed by 34 cycles at 94 °C for 30 s, 50 °C for 1 min and 72 °C for 1.5 min, and final extension at 72 °C for 10 min. DNA sequencing was performed at Beijing Genomics Institute, and the newly generated sequences were deposited in GenBank (Table 1). BioEdit v.7.0.5.3 [20] was used to review the chromatograms and for contig assembly.

### 2.3. Phylogenetic Analyses

The molecular phylogenetic analyses were inferred from two separate concatenated ITS-28S sequences datasets of species in the Peniophoraceae and *Peniophora* s.l., respectively. *Amylostereum chailletii* (Pers.) Boidin was elected as the outgroup for the Peniophoraceae dataset, whilst *Confertobasidium olivaceoalbum* (Bourdot & Galzin) Jülich and *Metulodontia nivea* (P. Karst.) Parmasto were used for the *Peniophora* s.l. dataset. The ITS and 28S sequences were aligned separately using MAFFT v.7 [21] with the G-INS-I iterative refinement algorithm and optimized manually in BioEdit v.7.0.5.3. The separate alignments were then concatenated using Mesquite v.3.5.1 [22].

Maximum parsimony (MP), maximum likelihood (ML) analyses and Bayesian inference (BI) were carried out by using PAUP* v.4.0b10 [23], RAxML v.8.2.10 [24] and MrBayes 3.2.6 [25], respectively. In MP analysis, trees were generated using 100 replicates of the random stepwise addition of sequence and tree-bisection reconnection (TBR) branch-swapping algorithm with all characters given equal weight. Branch supports for all parsimony analyses were estimated by performing 1000 bootstrap replicates with a heuristic search of 10 random-addition replicates for each bootstrap replicate. In ML analysis, statistical support values were obtained using rapid bootstrapping with 1000 replicates, with default settings used for other parameters. For BI, the best-fit substitution model was estimated with jModeltest v.2.17 [26]. Four Markov chains were run for 0.3 and 20 million generations for the Peniophoraceae and *Peniophora* s.l. datasets, respectively, until the split deviation frequency values were lower than 0.01. Trees were sampled every 100 generations. The first quarter of the trees, which represented the burn-in phase of the analyses, were discarded, and the remaining trees were used to calculate posterior probabilities (BPP) in the majority rule consensus tree.

## 3. Results

### 3.1. Phylogenetic Analyses

Thirty-nine ITS and 39 28S sequences were newly generated in this study (Table 1). The Peniophoraceae dataset contained 33 ITS and 34 28S sequences from 34 samples representing 13 genera and the outgroup, and it had an aligned length of 2313 characters, of which 544 were parsimony-informative. MP analysis yielded 5000 equally parsimonious trees (TL = 2605, CI = 0.528, RI = 0.568, RC = 0.300, HI = 0.472). The *Peniophora* s.l. dataset contained 113 ITS and 97 28S sequences from 114 samples representing 63 taxa and the outgroup, and it had an aligned length of 1993 characters, of which 350 were parsimony-informative. MP analysis yielded 5000 equally parsimonious trees (TL = 1973, CI = 0.372, RI = 0.698, RC = 0.260, HI = 0.628). The jModelTest suggested GTR+I+G as the best-fit model of nucleotide evolution for both Peniophoraceae and *Peniophora* s.l. datasets. The average standard deviation of split frequencies of BI was 0.004389 (for the Peniophoraceae dataset) and 0.004179 (for the *Peniophora* s.l. dataset) at the end of the run. ML and BI analyses resulted in almost identical tree topologies compared to the MP analysis. The MP trees of Penophoraceae and *Peniophora* s.l. are shown in Figure 1 and Figure 2, respectively, with the parsimony bootstrap values (≥50%, front), likelihood bootstrap values (≥50%, middle) and Bayesian posterior probabilities (≥0.95, back) labeled along the branches.

**Table 1 jof-09-00093-t001:** Species and sequences used in the phylogenetic analyses. New species are set in bold with type specimens indicated with an asterisk (*).

Taxa	Voucher	Locality	ITS	28S	Reference
*Amylostereum chailletii*	NH8031/1035	Romania	AF506406	AF506406	[6]
*Asterostroma laxum*	EL33-99	Estonia	AF506410	AF506410	[6]
*A. muscicola*	KHL9537	Puerto Rico	AF506409	AF506409	[6]
*Baltazaria eurasiaticogalactina*	CBS666.84	France	—	AY293211	[27]
*B. octopodites*	FLOR 56449	Brazil	MH260025	MH260047	[28]
*Confertobasidium olivaceoalbum*	FP90196	USA	AF511648	AF511648	[6]
*Dichostereum effuscatum*	GG930915	France	AF506390	AF506390	[6]
*D. pallescens*	NH7046/673	Canada	AF506392	AF506392	[6]
*Gloiothele lactescens*	EL8-98	Sweden	AF506453	AF506453	[6]
*G. lamellosa*	KHL11031	Venezuela	AF506454	AF506454	[6]
*Lachnocladium schweinfurthianum*	KM 49740	Cameroon	MH260033	MH260051	[28]
*L. sp.*	KHL10556	Jamaica	AF506461	AF506461	[6]
*Metulodontia nivea*	NH13108/2712	Russia	AF506423	AF506423	[6]
*Peniophora albobadia*	CBS329.66	France	MH858809	MH870448	[9]
*P. albobadia*	He2159	USA	MK588755	MK588795	Present study
*P. aurantiaca*	UBCF:19732	—	HQ604854	HQ604854	—
*P. aurantiaca*	CBS396.50	France	MH856678	MH868195	[9]
*P. bicornis*	He3609	China	MK588763	MK588803	Present study
*P. bicornis*	He4767	China	MK588764	MK588804	Present study
*P. borbonica*	He4606	China	MK588765	MK588805	Present study
*P. borbonica*	He4597	China	MK588766	MK588806	Present study
*P. cinerea*	NH9808/1788	Spain	AF506424	AF506424	[6]
*P. cinerea*	CBS261.37	Belgium	MH855905	MH867412	[9]
*P. cinerea*	He3725	China	MK588769	MK588809	Present study
*P. crassitunicata*	CBS663.91	Reunion	MH862292	MH873972	[9]
*P. crassitunicata*	He3814	China	MK588770	MK588810	Present study
*P. cremicolor*	**He5380 ***	**China**	**MK588791**	**MK588831**	Present study
*P. duplex*	TPDuB1022	—	AF119519	—	[29]
*P. duplex*	CBS286.58	Canada	MH857787	MH869321	[9]
*P. erikssonii*	CBS287.58	France	MH857788	MH869322	[9]
*P. erikssonii*	Cui11871	China	MK588771	MK588811	Present study
*P. exima*	T-523	USA	MK588772	MK588812	Present study
*P. fasticata*	CBS942.96	Ethiopia	MH862624	—	[9]
*P. fissilis*	CBS681.91	Reunion	MH862298	MH873975	[9]
*P. fissilis*	CBS684.91	Mascarene Islands	MH862299	MH873976	[9]
*P. gilbertsonii*	CBS357.95	USA	MH862528	MH874164	[9]
*P. gilbertsonii*	CBS360.95	USA	MH862530	MH874165	[9]
*P. guadelupensis*	CBS715.91	Guadeloupe	MH862304	MH873977	[9]
*P. halimi*	CBS863.84	France	MH861844	MH873532	[9]
*P. halimi*	CBS864.84	France	MH861845	MH873533	[9]
*P. incarnata*	NH10271/1909	Danmark	AF506425	AF506425	[6]
*P. incarnata*	CBS399.50	France	MH856681	MH868198	[9]
*P. junipericola*	CBS349.54	Sweden	MH857354	—	[9]
*P. junipericola*	He2462	China	MK588773	MK588813	Present study
*P. kuehneri*	CBS719.91	Mascarene Islands	MH862307	MH873980	[9]
*P. kuehneri*	He4745	China	MK588757	MK588797	Present study
*P. kuehneroides*	CBS731.91	Mascarene Islands	MH862317	MH873989	[9]
*P. kuehneroides*	CBS732.91	Mascarene Islands	MH862318	MH873990	[9]
*P. laete*	CBS256.56	France	MH857617	MH869165	[9]
*P. laete*	FCUG 2681	Russia	GU322869	—	[30]
*P. lassa*	SP6129	Russia	KJ509191	—	[31]
*P. lassa*	He3052	China	MK588758	MK588798	Present study
*P. lassa*	Dai17081A	China	MK588759	MK588799	Present study
*P. laurentii*	CBS325.73	Norway	—	MH872397	[9]
*P. laxitexta*	BAFC 3309	Argentina	FJ882040	—	[32]
*P. laxitexta*	LGMF1159	Brazil	JX559580	—	[33]
*P. lilacea*	CBS337.66	Armenia	MH858813	MH870452	[9]
*P. limitata*	olrim963	Lithuania	AY787678	—	[34]
*P. lycii*	CBS264.56	France	MH857624	MH869169	[9]
*P. lycii*	Boid-437	France	MK588774	MK588814	Present study
*P. major*	**He5528 ***	**China**	**MK588792**	**MK588832**	Present study
*P. malaiensis*	CBS679.91	Singapore	MH862297	MH873974	[9]
*P. manshurica*	He2956	China	MK588776	MK588816	Present study
*P. manshurica*	He3729	China	MK588777	MK588817	Present study
*P. meridionalis*	CBS289.58	France	MH857789	MH869323	[9]
*P. molesta*	CBS677.91	Gabon	MH862295	—	[9]
*P. molesta*	CBS676.91	Gabon	MH862294	MH873973	[9]
*P. monticola*	CBS649.91	Reunion	MH862289	MH873970	[9]
*P. nuda*	He5280	China	MK588778	MK588818	Present study
*P. nuda*	HHB-4916-Sp	USA	MK588779	MK588819	Present study
*P. ovalispora*	CBS653.91	Mascarene Islands	MH862290	MH873971	[9]
*P. parvocystidiata*	CBS716.91	Guadeloupe	MH862305	MH873978	[9]
*P. piceae*	olrim10	Sweden	AY781264	—	[35]
*P. piceae*	209	Russia	JX507718	—	[36]
*P. pilatiana*	CBS265.56	France	MH857625	MH869170	[9]
*P. pilatiana*	CBS-A1/A2	—	MK588780	MK588820	Present study
*P. pini*	Hjm 18143	Sweden	EU118651	EU118651	[7]
*P. pini*	CBS274.56	France	MH857632	MH869177	[9]
*P. pithya*	CBS277.56	France	MH857635	MH869180	[9]
*P. pithya*	He3107	China	MK588781	MK588821	Present study
*P. polygonia*	CBS404.50	France	MH856684	MH868201	[9]
*P. polygonia*	He4651	China	MK588782	MK588822	Present study
*P. proxima*	CBS405.50	France	MH856685	MH868202	[9]
*P. proxima*	He5498	China	MK588783	MK588823	Present study
*P. pseudopini*	TPPpB1007	—	AF119514	—	[29]
*P. pseudopini*	DAOM-30124-Sp	Canada	MK588784	MK588824	Present study
*P. pseudoversicolor*	CBS338.66	France	MH858814	MH870453	[9]
*P. pseudoversicolor*	He5132	China	MK588785	MK588825	Present study
*P. quercina*	CBS407.50	France	MH856687	MH868204	[9]
*P. quercina*	CBS408.50	France	MH856688	MH868205	[9]
*P. reidii*	CBS397.83	France	MH861616	MH873334	[9]
*P. rhoica*	CBS943.96	Ethiopia	MH862625	MH874246	[9]
*P. roseoalba*	CLZhao3513	China	ON786559	OP380690	[16]
*P. roseoalba*	CLZhao9401	China	ON786560	—	[16]
*P. roseoalba*	He5031	China	OP872571	OP872575	Present study
*P. rufa*	CBS351.59	Canada	MH857891	MH869432	[9]
*P. rufa*	He2788	China	MK588786	MK588826	Present study
*P. rufomarginata*	CBS281.56	France	MH857639	MH869183	[9]
*P. rufomarginata*	CBS282.56	France	MH857640	MH869184	[9]
*P. septentrionalis*	CBS294.58	Canada	MH857791	MH869325	[9]
*P. shenghuae*	**He3507 ***	**China**	**MK588788**	**MK588828**	Present study
*P. shenghuae*	**He3535**	**China**	**MK588789**	**MK588829**	Present study
*P. shenghuae*	**He5447**	**China**	**MK588790**	**MK588830**	Present study
*P. simulans*	CBS874.84	France	MH861849	MH873537	[9]
*P. simulans*	CBS875.84	France	MH861850	MH873538	[9]
*P. sphaerocystidiata*	HHB-8827-Sp	USA	MK588787	MK588827	Present study
*P. subsalmonea*	CBS696.91	Mascarene Islands	MH862302	—	[9]
*P. subsalmonea*	CBS697.91	Mascarene Islands	MH862303	—	[9]
*P. taiwanensis*	He4870	China	MK588775	MK588815	Present study
*P. taiwanensis*	Wu 9206-28	China	MK588793	MK588833	Present study
*P. taiwanensis*	Wu 9209-14	China	MK588794	MK588834	Present study
*P. tamaricicola*	CBS438.62	Morocco	MH858203	MH869802	[9]
*P. tamaricicola*	CBS439.62	Morocco	MH858204	MH869803	[9]
*P. trigonosperma*	CBS402.83	France	MH861618	MH873335	[9]
*P. trigonosperma*	He3602	China	MK588762	MK588802	Present study
*P. tristicula*	CBS210.63	Pakistan	MH858266	—	[9]
*P. tristicula*	He4775	China	MH669235	MH669239	[37]
*P. versicolor*	CBS358.61	Morocco	MH858082	MH869651	[9]
*P. versiformis*	CBS358.54	France	MH857360	MH868902	[9]
*P. versiformis*	He3029	China	MK588756	MK588796	Present study
*P. vietnamensis*	**He5242**	**Vietnam**	**MK588760**	**MK588800**	Present study
*P. vietnamensis*	**He5252 ***	**Vietnam**	**MK588761**	**MK588801**	Present study
*P. violaceolivida*	CBS348.52	France	MH857077	MH868613	[9]
*P. yunnanensi*	CLZhao3978	China	OP380617	OP380689	[16]
*P. yunnanensi*	CLZhao7347	China	OP380616	—	[16]
*Scytinostroma jacksonii*	NH6626/635	Canada	AF506467	AF506467	[6]
*S. portentosum*	EL11-99	Sweden	AF506470	AF506470	[6]
*S. renisporum*	CBS770.86	Indonesia	MH862050	MH873737	[9]
*Vararia investiens*	TAA164122	Norway	AF506484	AF506484	[6]
*V. ochroleuca*	JS24400	Norway	AF506485	AF506485	[6]
*Vesiculomyces citrinus*	EL53-97	Sweden	AF506486	AF506486	[6]

In the trees, species of *Peniophora* s.l. including the type species of *Peniophora* s.s., *Dendrophora* and *Duportella* formed a strongly supported clade (95/95/1 in Figure 1, 100/100/1 in Figure 2). Four distinct lineages corresponding to *Peniophora cremicolor*, *P. major*, *P. shenghuae* and *P. vietnamensis* spp. nov. were recovered. For other sequences generated in this study, they formed distinct lineages alone or together with sequences from GenBank and represented known species.

### 3.2. Taxonomy

***Peniophora*** Cooke, Grevillea 8 (no. 45): 20, 1879, *emended*

Basidiomes annual or perennial, resupinate, effused or effused-reflexed, adnate or sometimes loosening with age, membranaceous, ceraceous or coriaceous, usually stratified in section. Reflexed parts narrow, velutinous, felty to tomentose. Hymenophore smooth or tuberculate, rarely raduloid or merulioid, reddish, orange, pink, violaceous, gray, cream-colored, ochraceous, brown to dark brown; margin indistinct to fibrillose, adnate or slightly elevated, usually curved inside in reflexed parts when dry and old, concolorous or darker than hymenophore surface. Hyphal system monomitic or dimitic. Skeletal hyphae thick-walled, yellowish brown, dominated in the subiculum. Generative hyphae colorless to yellowish brown, thin- to thick-walled, with or without clamps. Dendrohyphidia present in some species, colorless or brown, thin- or thick-walled. Lamprocystidia present in most species, subulate or subcylindrical, colorless to brown, mostly thick-walled, encrusted with crystals. Gloeocystidia present or not, fusiform, subclavate to subcylindrical, colorless, thin- to thick-walled, empty or with contents. Basidia clavate to subcylindrical, sometimes flexuous, thin- to slightly thick-walled, with (2–) 4 sterigmata. Basidiospores colorless, usually cylindrical, allantoid or ellipsoid, rarely ovoid, subglobose, lacrimoid, pyriform or triangular, thin-walled, smooth, inamyloid, acyanophilous, spore print mostly pink.

Notes—According to the phylogenetic analyses, the morphological characters used to delimit the three genera, *Peniophora s.s.*, *Dendrophora* and *Duportella*, are not reliable. Thus, we propose to use a broad concept for *Peniophora*, which includes taxa possessing resupinate to effused-reflexed basidiomes with a bright or dull hymenophore surface, a monomitic or dimitic hyphal system with simple-septate or clamped generative hyphae, colorless or brown lamprocystidia and differently shaped basidiospores. However, we believe that the simultaneous presences of lamprocystidia and gloeocystidia, and the inamyloid basidiospores can be regarded as the synapomorphic traits for *Peniophora* in Russulales.

***Peniophora cremicolor*** Y.L. Xu, Y. Tian & S.H. He, **sp. nov.** Figure 3

MycoBank: MB846853

Type—CHINA, Fujian Province, Wuyishan County, Wuyishan Nature Reserve, on dead but still attached branch of angiosperm tree, 6 April 2018, He 5380 (BJFC 026441, holotype).

Etymology—refers to the species having the cream color of the dried basidiomes.

Fruiting body—Basidiomes annual, resupinate, effused, adnate or slightly detached from substrate with age, membranaceous to coriaceous, first as small colonies, later confluent up to 20 cm long, 2 cm wide, 200 µm thick in section. Hymenophore smooth, orange yellow [4B(7–8)] when fresh, cream [4A3] to grayish yellow [4B3] after dry, not cracked, unchanged in KOH; margin thinning out, fimbriate and white when juvenile, becoming indistinct and concolorous with hymenophore surface when mature.

Microscopic structures—Hyphal system monomitic; generative hyphae simple-septate. Subiculum well-developed, colorless, with a compact texture; hyphae colorless, thin- to slightly thick-walled, smooth, rarely branched, moderately septate, agglutinated, more or less parallel to substrate, 2–4 µm in diam. Subhymenium thickening with age, composed of lamprocystidia, hyphae and collapsed hymenial elements; hyphae colorless, thin- to slightly thick-walled, smooth, agglutinated, interwoven, 2–3.5 µm in diam. Lamprocystidia numerous, metuloid, subulate, colorless, thick-walled, heavily encrusted with crystals in the middle and upper parts, embedded, with a basal simple septum, 32–70 × 6–12 μm (with encrustations). Gloeocystidia subcylindrical to subclavate, usually with one or two contractions, empty or sometimes with contents, colorless, slightly thick-walled in most parts, slightly thickening toward the base, with a basal simple septum, smooth, 50–80 × 7–12 μm. Basidia clavate, colorless, slightly thick-walled, with a basal simple septum and four sterigmata, usually projecting beyond the hymenium, 45–65 × 8–12 μm; basidioles numerous, in shape similar to basidia but slightly smaller. Basidiospores ellipsoid to broadly ellipsoid, colorless, thin-walled, smooth, bearing a distinct apiculus, inamyloid, acyanophilous, 9–13 × 6–7 µm, L = 11.3 µm, W = 6.3 µm, Q = 1.8 (n = 60/2).

Additional specimen examined—CHINA, Fujian Province, Wuyishan County, Wuyishan Nature Reserve, on dead but still attached branch of angiosperm tree, 6 April 2018, He 5385 (BJFC 026446).

Notes—*Peniophora cremicolor* is characterized by the cream-colored basidiomes, simple-septate generative hyphae, numerous lamprocystidia and gloeocystidia, large ellipsoid to broadly ellipsoid basidiospores. In the phylogenetic tree (Figure 2), *P. cremicolor* formed a distinct lineage sister to *P. fasticata* Boidin & Lanq., *P. subsalmonea* Boidin, Lanq. & Gilles and *P. rhoica*, all of which have cylindrical to allantoid basidiospores [5]. Morphologically, *P. erikssonii* Boidin is similar to *P. cremicolor* by having orange yellow basidiomes, simple-septate generative hyphae and ellipsoid basidiospores, but differs in having larger lamprocystidia (50–110 × 8–15 µm), gloeocystidia (70–140 × 10–15 µm) and basidiospores (13–20 × 8–13 µm) [5]. *Peniophora proxima* Bres. is similar to *P. cremicolor* by sharing same size of basidiospores but differs in having clamped generative hyphae and smaller lamprocystidia (15–40 × 5–7 µm) [38].

***Peniophora major*** Y.L. Xu, Y. Tian & S.H. He, **sp. nov.** Figure 4

MycoBank: MB846854

Type—CHINA, Guizhou Province, Jiangkou County, Fanjingshan Nature Reserve, on dead angiosperm branch, 11 July 2018, He 5528 (BJFC 026589, holotype).

Etymology—refers to the species having large lamprocystidia, gloeocystidia and basidiospores.

Fruiting body—Basidiomes annual, resupinate, effused, closely adnate, inseparable from substrate, coriaceous, up to 35 cm long, 3 cm wide, 350 µm thick in section. Hymenophore smooth, pale orange [6A3], orange grey [6B2] to grayish orange [6B(3–4)], not cracked, unchanged in KOH; margin thinning out, indistinct, concolorous with hymenophore surface.

Microscopic structures—Hyphal system monomitic; generative hyphae simple-septate. Subiculum thin, yellowish brown, with a compact texture; hyphae colorless to pale yellow, thick-walled, smooth, moderately septate, rarely branched, densely agglutinated, more or less parallel to substrate, 3–5 µm in diam. Subhymenium thickening, composed of lamprocystidia, hyphae and collapsed hymenial elements; hyphae colorless, thin- to slightly thick-walled, smooth, agglutinated, interwoven, 2–4 µm in diam. Lamprocystidia numerous, metuloid, subulate, colorless or pale yellow at base with age, thick-walled, heavily encrusted with crystals in the middle and upper parts, embedded or projecting beyond the hymenium, with a basal simple septum, 40–105 × 11–16 μm (with encrustations). Gloeocystidia subclavate to subcylindrical, colorless, slightly thick-walled, smooth, often with contents, with a basal simple septum, 40–85 × 7–10 μm. Basidia subclavate, usually with a contraction in the upper part, colorless, slightly thick-walled, with a basal simple septum and four sterigmata, 35–50 × 8–9 μm; basidioles numerous, in shape similar to basidia but slightly smaller. Basidiospores cylindrical, usually slightly concaved in one side, bearing a distinct apiculus, colorless, thin-walled, smooth, inamyloid, acyanophilous, 10–14 × 4.8–6 µm, L = 12.2 µm, W = 5.1 µm, Q = 2.4 (*n* = 30/1).

Notes—*Peniophora major* is characterized by the grayish basidiomes, simple-septate generative hyphae, and large lamprocystidia, gloeocystidia and basidiospores. In the phylogenetic tree (Figure 2), *P. major* formed a sister lineage to *P. limitata* (Chaillet ex Fr.) Cooke, but their relationship was not well-supported. Morphologically, *P. limitata* differs from *P. major* by clamped generative hyphae, the absence of gloeocystidia, and smaller lamprocystidia (25–60 × 8–12 µm) and basidiospores (7.5–12 × 2.5–3.5 µm) [5]. *Peniophora coprosmae* G. Cunn. from Australia and New Zealand is similar to *P. major* by sharing distinct gloeocystidia and large cylindrical basidiospores, but it differs in having clamped generative hyphae, slightly shorter lamprocystidia (35–80 µm long) and slightly smaller basidiospores (9–11.5 × 4–5 µm) [5].

***Peniophora shenghuae*** Y.L. Xu, Y. Tian & S.H. He, **sp. nov.** Figure 5

MycoBank: MB846855

Type—CHINA, Yunnan Province, Luquan County, Zhuanlong Town, on dead angiosperm branch, 4 December 2015, He 3507 (BJFC 021904, holotype).

Etymology—Named to honor Dr. Sheng-Hua Wu (National Museum of Natural Science, Taiwan) who contributed to the taxonomy of *Peniophora* s.l. and other corticioid fungi in Taiwan.

Fruiting body—Basidiomes annual, resupinate, effused, closely adnate, coriaceous, first as small colonies, later confluent up to 9 cm long, 3 cm wide, 200 µm thick in section. Hymenophore smooth, pale orange [5A3], orange gray [5B2] to brownish orange [5C (3–6)], not cracked or with scattered crevices when dry, slightly darkening in KOH; margin thinning out, indistinct, usually darker than hymenophore surface, brown.

Microscopic structures—Hyphal system monomitic; generative hyphae simple-septate. Subiculum yellowish brown, with a compact texture, up to 60 µm thick; hyphae colorless to pale yellow, thin- to slightly thick-walled, smooth, rarely branched, moderately septate, interwoven, densely agglutinated, 2–4 µm in diam. Subhymenium thickening, composed of lamprocystidia, hyphae and collapsed hymenial elements; hyphae colorless, thin- to slightly thick-walled, smooth, tightly interwoven, 2–3 µm in diam. Lamprocystidia abundant, subulate, colorless, thick-walled, heavily encrusted with crystals in most parts, mostly embedded or slightly projecting beyond the hymenium, with a basal simple septum, 20–60 × 8–16 μm (with encrustations). Gloeocystidia fusiform, usually with a small apical papilla, flexuous, colorless, slightly thick-walled, smooth, empty or with content, 28–50 × 5–8 μm. Basidia subclavate to subcylindrical, usually with several contractions, flexuous, colorless, thin-walled, smooth, with a basal simple septum and four sterigmata, 24–45 × 5–7 μm; basidioles numerous, in shape similar to basidia but slightly smaller. Basidiospores cylindrical, bearing a distinct apiculus, colorless, thin-walled, smooth, inamyloid, acyanophilous, 7.5–10 (–10.5) × 3–4 µm, L = 9.1 µm, W = 3.3 µm, Q = 2.8 (*n* = 30/1).

Additional specimens examined—CHINA, Yunnan Province, Luquan County, Zhuanlong Town, on dead angiosperm branch, 4 December 2015, He 3506 (BJFC 021903) & He 3535 (BJFC 021933); Guizhou Province, Chishui County, Suoluo Nature Reserve, on dead liana, 7 July 2018, He 5447 (BJFC 026508).

Notes—*Peniophora shenghuae* is characterized by the grayish orange basidiomes, simple-septate generative hyphae, numerous strongly encrusted lamprocystidia, papillated gloeocystidia and cylindrical basidiospores. In the phylogenetic tree (Figure 2), *P. shenghuae* formed a strongly supported lineage with *P. reidii* Boidin & Lanq. Meanwhile, morphologically, *P. reidii* is similar to *P. shenghuae* by sharing simple-septate generative hyphae, papillated gloeocystidia and same size, cylindrical basidiospores. However, *P. reidii* has thicker basidiomes (up to 0.6 mm) and wider, sometimes bifurcate lamprocystidia (12–20 μm wide) [5]. The occurrence of *P. reidii* in Taiwan needs to be confirmed [11]. *Peniophora borbonica* Boidin & Gilles is similar to *P. shenghuae* by sharing simple-septate generative hyphae, fusiform gloeocystidia and cylindrical basidiospores, but it differs in having thicker basidiomes (up to 1.2 mm thick), slightly thick-walled basidia and slightly slender basidiospores (2.7–3.5 μm wide) [5]. Two specimens collected from Taiwan were identified as *P. borbonica* by us, which formed a distinct lineage distantly related to the lineage of *P. shenghuae* and *P. reidii* (Figure 2).

***Peniophora vietnamensis*** Y.L. Xu, Y. Tian & S.H. He, **sp. nov.** Figure 6

MycoBank: MB846856

Type—VIETNAM, Lam Dong Province, Bi Doup Nui Ba National Park, on dead Araceae plant, 15 October 2017, He 5252 (BJFC 024770, holotype).

Etymology—refers to the species being found in Vietnam.

Fruiting body—Basidiomes annual, resupinate, effused, closely adnate, first as small colonies, later confluent up to 9 cm long, 1 cm wide, 260 µm thick in section. Hymenophore smooth, brownish orange [6C(5–7)] to light brown [6D(6–8)], not cracked or finely cracked with age, turning reddish brown in KOH; margin thinning out, indistinct, concolorous or darker than hymenophore surface.

Microscopic structures—Hyphal system monomitic; generative hyphae simple-septate. Subiculum indistinct or absent; hyphae colorless, thin- to slightly thick-walled, smooth, rarely branched, moderately septate, interwoven, 2–3 µm in diam. Subhymenium thickening, with numerous lamprocystidia; hyphae colorless, thin- to slightly thick-walled, smooth, interwoven, 1.5–2.5 µm in diam. Lamprocystidia numerous, subulate, usually bifurcate, with one or two secondary septa, yellowish brown, distinctly thick-walled, slightly encrusted with fine crystals in the apex, embedded or slightly projecting beyond the hymenium, 25–40 × 5–10 μm. Gloeocystidia subulate, colorless, thin-walled, smooth, usually with one secondary septum, usually empty in the apical part, with contents in the basal part, 28–50 × 5–8 μm. Basidia subclavate, with a contraction in the middle part, colorless, thin-walled, with a basal simple septum and four sterigmata, 23–32 × 6–10 μm; basidioles numerous, in shape similar to basidia but slightly smaller. Basidiospores oblong cylindrical, colorless, thin- to slightly thick-walled, smooth, inamyloid, cyanophilous when thick-walled, 12–16.5 (–17) × 4–6 µm, L = 14.5 µm, W = 4.5 µm, Q = 3.2–3.3 (n = 60/2).

Additional specimen examined—VIETNAM, Lam Dong Province, Bi Doup Nui Ba National Park, on dead Araceae plant, 15 October 2017, He 5242 (BJFC 024760).

Notes—*Peniophora vietnamensis* is characterized by its brownish orange basidiomes on Araceae plant, simple-septate generative hyphae, brown lamprocystidia, subulate gloeocystidia and oblong cylindrical basidiospores. In the phylogenetic trees (Figure 1 and Figure 2), *P. vietnamensis* and *P. trigonosperma* Boidin, Lanq. & Gilles formed a strongly supported lineage. Both species grow on palm trees and have similar brown lamprocystidia, however, *P. trigonosperma* differs in having clamped hyphae, cylindrical to fusiform gloeocystidia and triangular basidiospores [5]. *Duportella rhoica* Boidin & Lanq. is similar to *P. vietnamensis* by sharing oblong cylindrical basidiospores (10–14.5 × 4–6 µm) but differs in having a dimitic hyphal system with skeletoid hyphae and clamped generative hyphae and larger fusiform gloeocystidia (60–90 × 5–12 µm) [5].

***Peniophora globispora*** Y.L. Xu & S.H. He, **nom. & comb. nov.**

MycoBank: MB846863

*Duportella sphaerospora* G. Cunn., Transactions and Proceedings of the Royal Society of New Zealand 85: 96 (1957).

Notes—*Duportella sphaerospora* is characterized by the large globose to subglobose basidiospores (10–12 × 9–12 µm) [5]. Since the epithet ‘*sphaerospora*’ was used in *Peniophora* for another species (*P. sphaerospora* Höhn. & Litsch. = *Hypochnicium sphaerosporum* (Höhn. & Litsch.) J. Erikss.), we propose the new name and new combination, *Peniophora globispora*, for this species.

***Peniophora jordaoensis*** (Hjortstam & Ryvarden) Y.L. Xu & S.H. He, **comb. nov.**

MycoBank: MB846857

*Duportella jordaoensis* Hjortstam & Ryvarden, Synopsis Fungorum 18: 20 (2004).

Notes—The species was originally described from Sao Paulo, Brazil, and it is known only from the type locality so far. Although sequences of the species are unavailable at present, morphologically, it has typical characteristics of *Peniophora*, such as the presence of both gloeocystidia and lamprocystidia, and suballantoid to allantoid basidiospores [39]. Meanwhile, the felted subiculum and brownish metuloid cystidia of the species can be seen in other species of *Peniophora*.

***Peniophora kuehneroides*** (Boidin, Lanq. & Gilles) Y.L. Xu & S.H. He, **comb. nov.**

MycoBank: MB846858

*Duportella kuehneroides* Boidin, Lanq. & Gilles, Bulletin de la Société Mycologique de France 107 (3): 98 (1991).

Notes—Two strains (CBS 731.91 and CBS 732.91) of *Duportella kuehneroides* from Mascarene Islands isolated and identified by Paule Lanquetin formed a distinct lineage in the large clade of *Peniophora* (Figure 1 and Figure 2). We propose the new combination based on both morphological and molecular evidence [5].

***Peniophora**lassa*** (Spirin & Kout) Y.L. Xu & S.H. He, **comb. nov.**

MycoBank: MB846859

*Duportella lassa* Spirin & Kout, Mycotaxon 130 (2): 484 (2015).

Notes—The species was recently described from northeast Russia [31]. A specimen from northeast China (He 3052) and the holotype of the species (Spirin 6129) formed a strongly supported lineage sister to other species of *Peniophora*. The close relationship between *Duportella lassa* and *Peniophora* was also noticed by Spirin and Kout [31]. *Peniophora lassa* is a common species in the Beijing area according to the authors’ investigations carried out recently.

Specimens examined—CHINA, Inner Mongolia Autonomous Region, Genhe County, Greater Khingan Mountains Nature Reserve, on trunk of *Salix*, 17 October 2015, He 3052 (BJFC 021442); Beijing, Mentougou District, Tanzhe Temple Park, on dead branch of *Syringa*, 17 July 2020, He 6511 (BJFC 033459); Jietai Temple Park, on dead branch of *Lonicera*, 7 August 2020, He 6735 (BJFC 033683); Pinggu District, Jinhai Lake Park, on trunk of *Cotinus*, 11 August 2020, He 6766 (BJFC 033714); Yanqing District, Songshan Nature Reserve, on dead branch of *Syringa*, 3 September 2020, He 6943 (BJFC 033892).

***Peniophora miranda*** (Boidin, Lanq. & Gilles) Y.L. Xu & S.H. He, **comb. nov.**

MycoBank: MB846860

*Duportella miranda* Boidin, Lanq. & Gilles, Bulletin de la Société Mycologique de France 107 (3): 100 (1991).

Notes—The species is characterized by the numerous brown lamprocystidia and subovoid basidiospores [3]. Boidin et al. (1998) showed that *P. miranda* was closely related to other *Peniophora* species, although the sequence is unavailable for us.

***Peniophora pirispora*** (Boidin, Lanq. & Gilles) Y.L. Xu & S.H. He, **comb. nov.**

MycoBank: MB846861

*Duportella pirispora* Boidin, Lanq. & Gilles, Bulletin de la Société Mycologique de France 107 (3): 104 (1991).

Notes—The species is characterized by the brown lamprocystidia and lacrimoid to pyriform or subreniform basidiospores. Although the sequences of the species are unavailable, present molecular evidence indicated that *Peniophora* includes species with differently shaped basidiospores (Figure 1 and Figure 2).

***Peniophora rhoica*** (Boidin & Lanq.) Y.L. Xu & S.H. He, **comb. nov.**

MycoBank: MB846862

*Duportella rhoica* Boidin & Lanq., Cryptogamie Mycologie 16 (2): 89 (1995).

Notes—In our phylogenetic trees (Figure 1 and Figure 2), one strain (CBS 943.96) isolated from the holotype of *Duportella rhoica* (LY 14759) was nested within the *Peniophora* clade. Thus, we propose the new combination based on molecular evidence and morphological descriptions and illustrations [40].

***Peniophora tristiculoides*** (Sheng H. Wu & Z.C. Chen) Y.L. Xu & S.H. He, **comb. nov.**

MycoBank: MB846864

*Duportella tristiculoides* Sheng H. Wu & Z.C. Chen, Bulletin of the National Museum of Natural Science 4: 108 (1993).

Notes—Morphologically, *Duportella tristiculoides* is close to *D. tristicula* (Berk. & Broome) Reinking, which was the type species of *Duportella* and nested within the *Peniophora* clade (Figure 1 and Figure 2). *Duportella tristiculoides* has been only reported from the type locality in Taiwan and not been sequenced so far.

***Peniophora malaiensis*** Boidin, Lanq. & Gilles, Bulletin de la Société Mycologique de France 107 (3): 137 (1991)

*Peniophora taiwanensis* Sheng H. Wu, Mycotaxon 85: 197 (2003)

Notes—Our phylogenetic analyses indicated that the holotype of *P. malaiensis*, LY 8292 from Singapore (CBS679.91), formed a strongly supported lineage with three specimens collected from Guangxi Autonomous Region and Taiwan, China, including the paratype of *P. taiwanensis*, Wu 9206-28 (Figure 2). The ITS sequence similarity between LY 8292 and Wu 9206-28 is 99.16% (5 base pair differences). Morphologically, the two species are very similar except that *P. taiwanensis* has narrower basidiospores (1.8–2.2 μm) than *P. malaiensis* (2–2.7 μm) [3,11]. However, our measurements of the basidiospore width of the specimen He 4870 match that of *P. malaiensis* (2–3 μm). Therefore, we believe that the difference in the basidiospore width is within the specific range and treat *P. taiwanensis* as a later synonym of *P. malaiensis*.

Specimen examined—CHINA, Guangxi Autonomous Region, Jinxiu County, Dayaoshan Nature Reserve, Shengtangshan, on fallen angiosperm trunk, 15 July 2015, He 4870 (BJFC 024389).

***Duportella renispora*** Boidin, Lanq. & Gilles, Bulletin de la Société Mycologique de France 107 (3): 104 (1991).

Notes—The strain (CBS 733.91) isolated from the holotype of the *D. renispora* (LY 12699) was sequenced by Vu et al. [9]. The blast results of the ITS (MH862319) and 28S (MH873991) sequences showed that the species does not belong to *Peniophora* s.l. but is close to *Amylostereum* Boidin. Morphologically, the thinly encrusted brown cystidia and absence of subiculum of the species do match the characteristics of *Amylostereum*, which, however, has amyloid basidiospores. In order to confirm the identity of the species, the type specimen should be checked in the future.
**A key to *Peniophora* species in China**
1. Dendrohyphidia well-developed, brown*P. versiformis*1. Dendrohyphidia absent, or colorless when present22. Lamprocystidia brown over the entire length32. Lamprocystidia colorless or only brown at the basal part63. Skeletocystidia present43. Skeletocystidia absent54. Basidiospores 10–12.5 × 3.7–4.2 µm*P. tristicula*4. Basidiospores 5.2–7.5 × 2–3.5 µm*P. kuehneri*5. Basidiospores triangular, on palms*P. trignosperma*5. Basidiospores cylindrical to narrowly ellipsoid or ovoid, on angiosperms*P. lassa*6. Dendrohyphidia present, often colorless and may be difficult to see76. Dendrohyphidia absent87. Lamprocystidia absent; basidiospores > 7.5 µm long*P. polygonia*7. Lamprocystidia present; basidiospores < 7.5 µm long*P. meridionalis*8. Basidiospores ellipsoid or ovoid98. Basidiospores cylindrical or allantoid129. Basidiospores ovoid*P. roseoalba*9. Basidiospores ellipsoid1010. Generative hyphae with clamps*P. proxima*10. Generative hyphae without clamps1111. Basidiospores 9–13 × 6–7 µm*P. cremicolor*11. Basidiospores 11–20 × 8–14 µm*P. erikssonii*12. Basidiomes red1312. Basidiomes grey, cream, pale orange, pinkish, yellowish or purplish1413. Basidiomes pulvinate, only found on *Populus**P. rufa*13. Basidiomes effused, found on other hosts*P. pseudoversicolor*14. Generative hyphae without clamps1514. Generative hyphae with clamps1915. Basidia with two sterigmata*P. bicornis*15. Basidia with four sterigmata1616. Basidiospores > 10 µm long*P. major*16. Basidiospores < 10 µm long1717. Basidiospores < 7 µm long*P. malaiensis*17. Basidiospores > 7 µm long1818. Basidiomes up to 1.2 mm thick; basidia thick-walled*P. borbonica*18. Basidiomes up to 200 µm thick; basidia thin-walled*P. shenghuae*19. Gloeocystidia indistinct or absent2019. Gloeocystidia present2220. Basidiomes pinkish, effused-reflexed, usually on *Quercus.**P. manshurica*20. Basidiomes greyish, resupinate, on gymnosperms or angiosperms2121. Lamprocystidia 40–80 × 6–14 µm; on *Juniperus**P. junipericola*21. Lamprocystidia 15–25 × 5–10 µm; on gymnosperms or angiosperms*P. cinerea*22. Gloeocystidia distinctly thick-walled, with walls up to 2–3 µm thick*P. crassitunicata*22. Gloeocystidia thin- to slightly thick-walled, with walls < 1 µm thick2323. Lamprocystidia up to 20 µm wide; basidiospores up to 7 µm long *P. pithya*23. Lamprocystidia up to 12 µm wide; basidiospores up to 10 µm long2424. Gloeocystidia 30–80 × 8–20 µm; basidiospores 2.5–3.5 µm wide*P. nuda*24. Gloeocystidia 12.5–58 × 5.5–15.5 µm; basidiospores 3–5.5 µm wide*P. yunnanensis*

## 4. Discussion

The modern molecular phylogenetics have significantly changed the taxonomic systems of wood-inhabiting fungi in recent years. On the one hand, many new lineages and taxa were found and described, and on another hand, some morphologically dissimilar taxa were proved to be closely related in phylogeny [41,42,43,44,45]. Our results indicated that in the *Peniophora* s.l. group, the morphological characters, such as the color of lamprocystidia and dendrohyphidia used for generic delimitation are not reliable. As already shown in other studies [46,47,48,49,50], the identities of some genera erected based merely on morphology or together with phylogenetic analyses of not well-sampled datasets need to be confirmed.

The Peniophoraceae is a well-supported large family in Russulales, most species of which have resupinate basidiomes and non-poroid hymenophores growing on twigs, branches or trunks of woody plants or bamboos [5,51]. The species diversity, taxonomy and phylogeny of Peniophoraceae in China have been investigated and studied [16,37,52,53,54,55], and this study is a part of this consecutive research. However, the two large genera in Peniophoraceae, *Scytinostroma* Donk and *Vararia* P. Karst., that are closely related and were shown to be polyphyletic [6,7,28], have not been sufficiently studied worldwide. We are going to carry out a complete morphological and phylogenetic analyses of the Peniophoraceae by adding a lot of *Scytinostroma*-*Vararia* samples from Asia accumulated by the authors in recent years.

## Figures and Tables

**Figure 1 jof-09-00093-f001:**
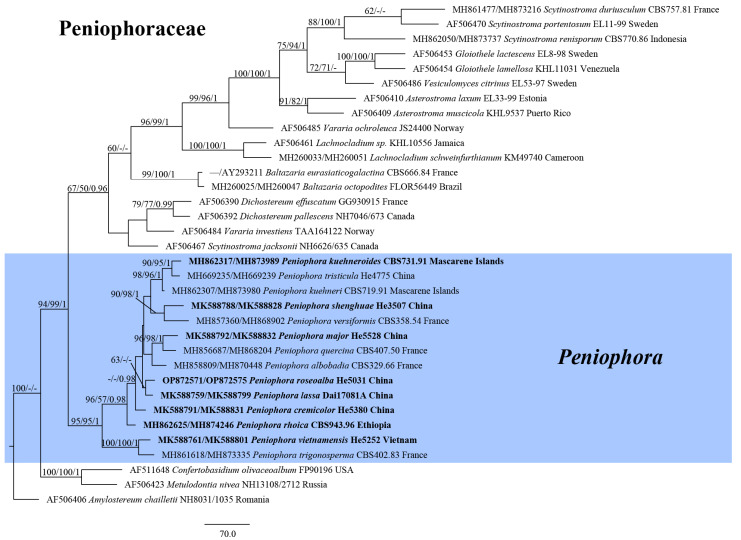
Phylogenetic tree obtained from maximum parsimony analysis of ITS-28S sequence data of Peniophoraceae. Branches are labeled with parsimony bootstrap values (≥50%, front), likelihood bootstrap values (≥50%, middle) and Bayesian posterior probabilities (≥0.95, back). New species and new combinations are set in bold.

**Figure 2 jof-09-00093-f002:**
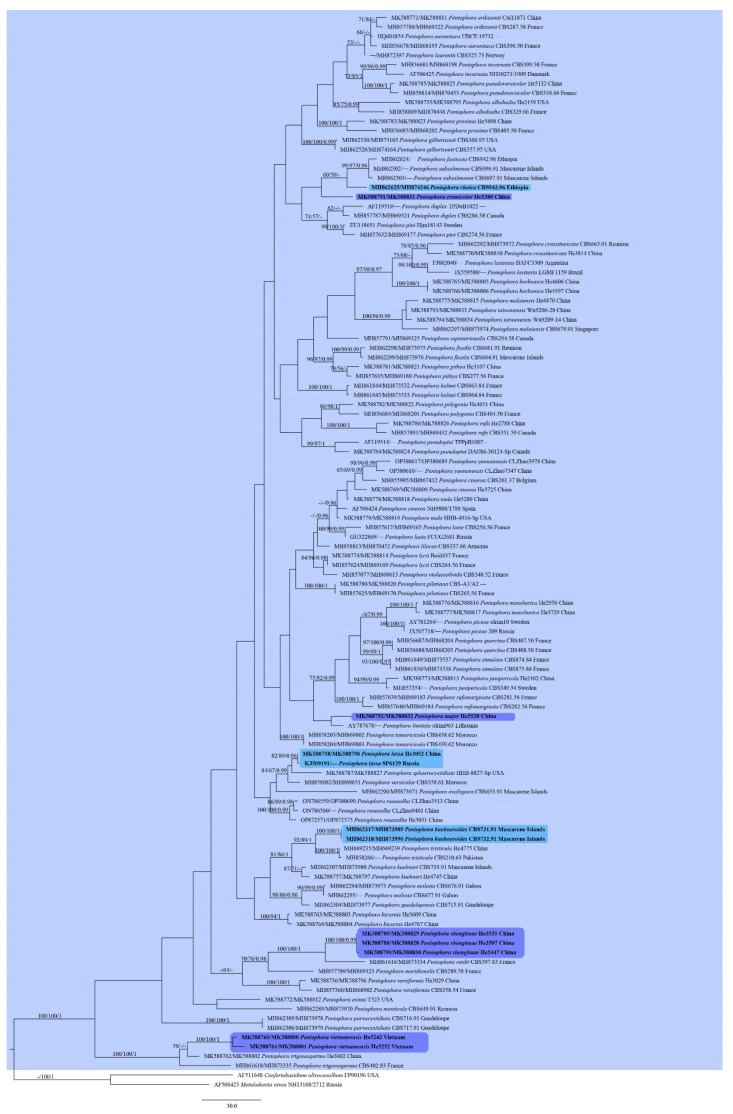
Phylogenetic tree obtained from maximum parsimony analysis of ITS-28S sequence data of *Peniophora* s.l. Branches are labeled with parsimony bootstrap values (≥50%, front), likelihood bootstrap values (≥50%, middle) and Bayesian posterior probabilities (≥0.95, back). New species (purple) and new combinations (blue) are set in bold.

**Figure 3 jof-09-00093-f003:**
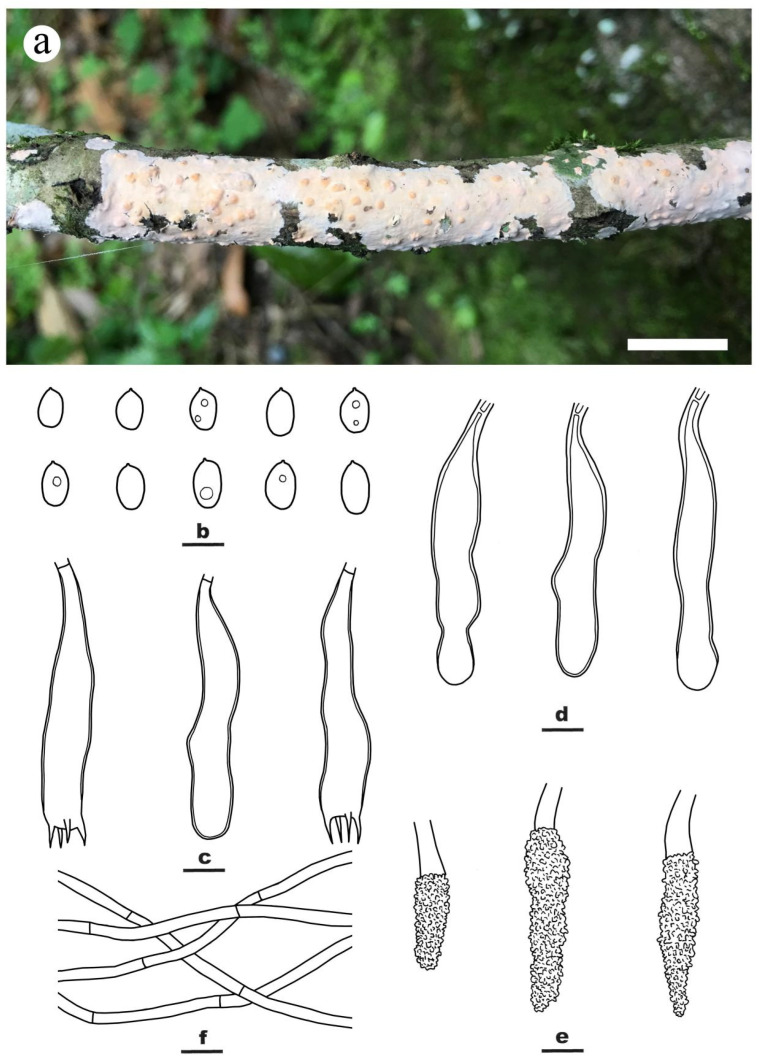
*Peniophora cremicolor* ((**a**) from paratype BJFC 026446; (**b–f**) from the holotype BJFC 026441). Scale bars: (**a**) = 1 cm; (**b**–**f**) = 10 µm. (**a**). Basidiomes; (**b**). Basidiospores; (**c**). Basidia and a basidiole; (**d**). Gloeocystidia; (**e**). Lamprocystidia; (**f**). Hyphae from subiculum.

**Figure 4 jof-09-00093-f004:**
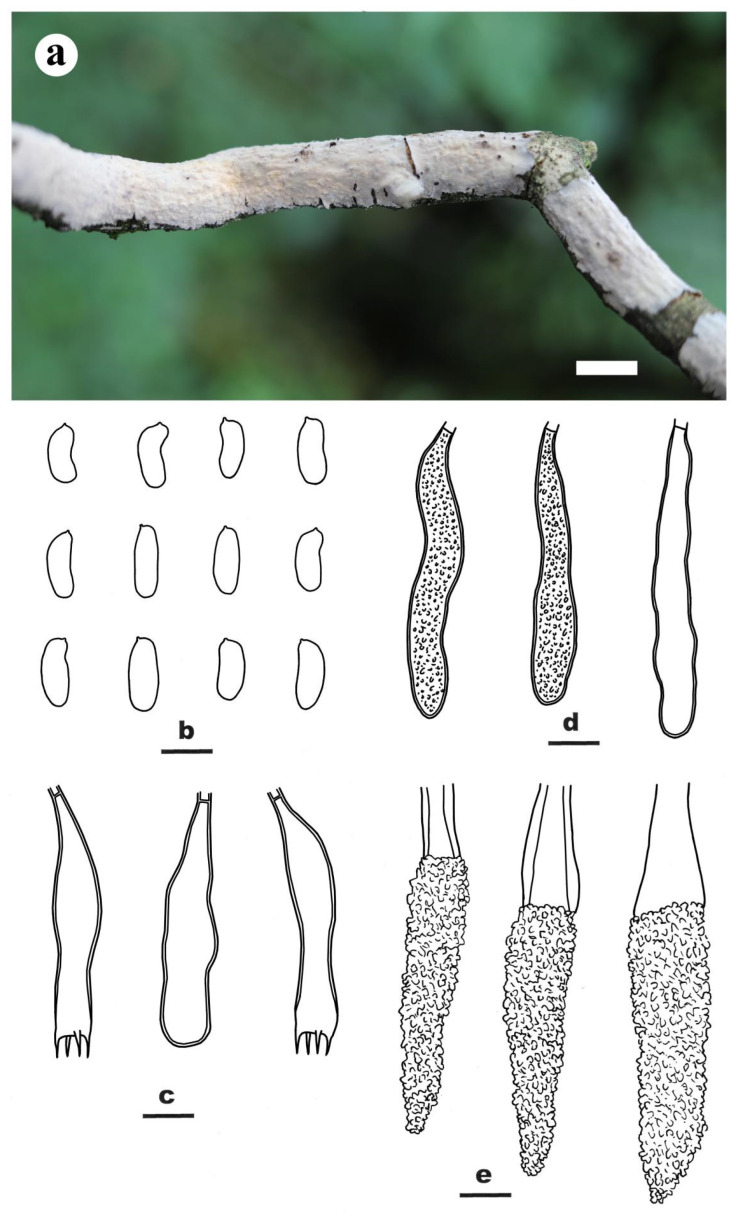
*Peniophora major* (from the holotype BJFC 026589). Scale bars: (**a**) = 1 cm; (**b**–**e**) = 10 µm. (**a**). Basidiomes; (**b**). Basidiospores; (**c**). Basidia and a basidiole; (**d**). Gloeocystidia; (**e**). Lamprocystidia.

**Figure 5 jof-09-00093-f005:**
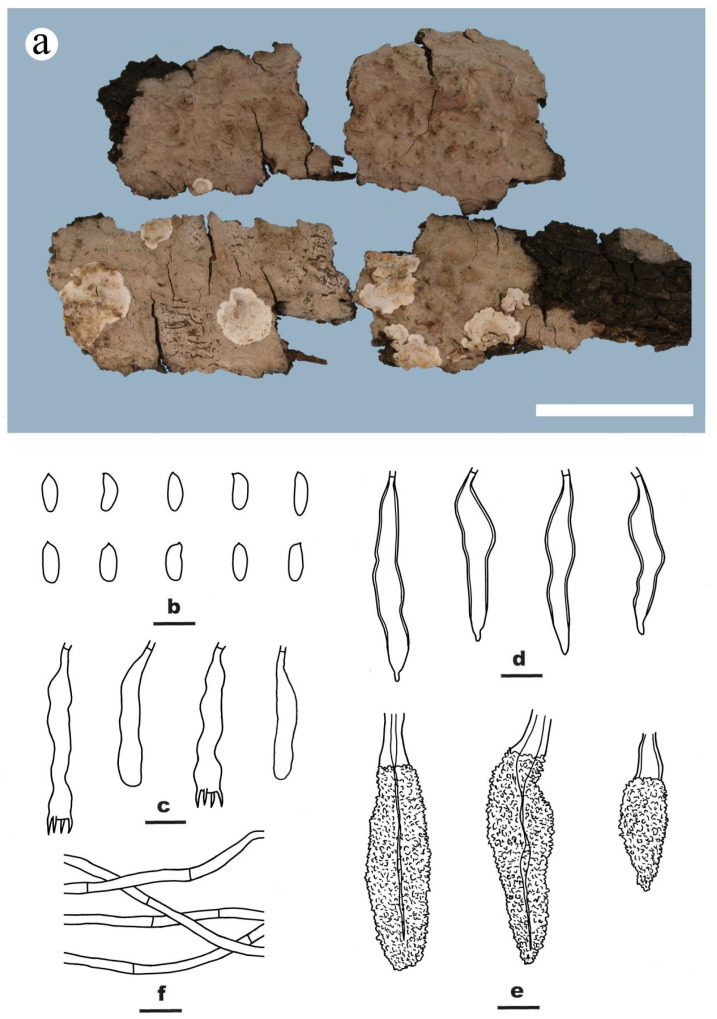
*Peniophora shenghuae* (from the holotype BJFC 021904). Scale bars: (**a**) = 1 cm; (**b**–**f**) = 10 µm. (**a**). Basidiomes; (**b**). Basidiospores; (**c**). Basidia and a basidiole; (**d**). Gloeocystidia; (**e**). Lamprocystidia; (**f**). Hyphae from subiculum.

**Figure 6 jof-09-00093-f006:**
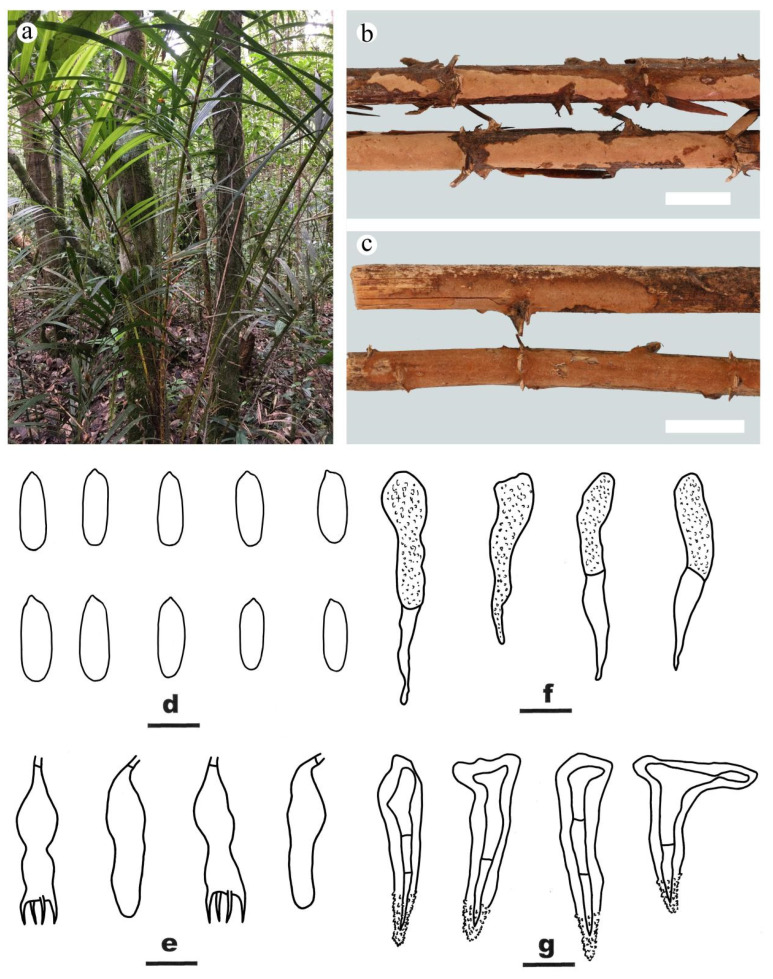
*Peniophora vietnamensis* ((**a**,**c**–**g**) from the holotype BJFC 024770; (**b**) from the paratype BJFC 024760). Scale bars: (**b**,**c**) = 1 cm; (**d**–**g**) = 10 µm. (**a**–**c**). Basidiomes; (**d**). Basidiospores; (**e**). Basidia and basidioles; (**f**). Gloeocystidia; (**g**). Lamprocystidia.

## Data Availability

Not applicable.

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
