# Peer review of "Taxonomy and Phylogeny of Peniophora Sensu Lato (Russulales, Basidiomycota)"

_jof, 2023, doi:10.3390/jof9010093_

Round 1

Reviewer 1 Report

Dear respected authors I suggested you to add a paragraph about the distribution and ecology of the genus Peniophora in the introduction before go into the taxonomy of the genus. Please check well the dichotomus key of the treated taxa.  

Please change this sentence in the introduction: The genus Peniophora Cooke (Peniophoraceae, Russulales), typified by Thelephora 33 quercina Pers. ex Fr., is one of the oldest genera of corticiod fungi. to:

The genus Peniophora Cooke (Peniophoraceae, Russulales) introduced in 1879, typified by Thelephora quercina Pers. ex Fr., is one of the oldest genera of corticiod fungi. 

Author Response

Dear respected reviewer, Thank you for the wonderful suggestions. Here are our responses: 

1. Dear respected authors I suggested you to add a paragraph about the distribution and ecology of the genus Peniophora in the introduction before go into the taxonomy of the genus. Please check well the dichotomus key of the treated taxa.  

R. We added the following two sentences at the begining of the introduction section: "It is a cosmopolitan genus widely distributed from boreal to tropical areas, causing white rots on both angiosperms and gymnosperms. Species of the genus prefer to grow on small branches especially those dead but still attached ones in the exposed and dry environments."

2. Please check well the dichotomus key of the treated taxa.

R. Yes, the authors double-checked the identification key to all the Peniophora species in China.

3. Please change this sentence in the introduction: The genus Peniophora Cooke (Peniophoraceae, Russulales), typified by Thelephora 33 quercina Pers. ex Fr., is one of the oldest genera of corticiod fungi. to:

The genus Peniophora Cooke (Peniophoraceae, Russulales) introduced in 1879, typified by Thelephora quercina Pers. ex Fr., is one of the oldest genera of corticiod fungi. 

R. Yes, we added the words "introduced in 1879" as suggested.

Reviewer 2 Report

The most important thing is that Peniophora macrospora has already been used for another species (Bresadola 1913). A new name will have to be devised.

The English standard and presentation is very high, which made it a pleasure to read this paper. I found just a few minor things that require changes. I detail them as follows.

line 14. Replace "sufficient" by "sufficiently"

line 16. Replace "of a" by "of" [i.e., delete "a"]

line 18. "were entangled" or "were interspersed" seems better than "entangled" 

line 27. Replace "the 25 species" by "25 species"

line 28. "further study" is better than "further studies"

line 33. "corticioid" not "corticiod"

line 35. "Index Fungorum" not "Index of Fungorum"

line 38. "has" is better than "was"

lines 6263. "explore" sounds better than "figure out" in a scientific publication.

line 122. "trees" not "tree"

line 125. "trees" not "tree"

line 153. "loosen"?? Not a mycological term; what exactly is meant here?

line 226. Peniophora macrospora ia an invalid name, as it is a homonym of P. macrospora Bres. 1913 [see Species Fungorum; a new name will need to be created.]

line 331. "slightly" not "slight"

line 365. "propose" not "proposed"

line 390. Delete "Asia of". That is, it should read "from northeast Russia"

line 394. I think "in the Beijing area" would read better.

line 419. "largely" seems superfluous and can be deleted.

line 425. Replace "were" by "was"

line 525. I would put a comma after "Russulales" and delete the comma after "well-supported"

Author Response

Dear respected reviewer, thank you for the wonderful revisions. Here are our responses:

1. The most important thing is that Peniophora macrospora has already been used for another species (Bresadola 1913). A new name will have to be devised.

R. We devised another name "Peniophora major" for our new species, and replaced "macrospora" with "major" in the phylogenetic trees and the whole text.

2. line 153. "loosen"?? Not a mycological term; what exactly is meant here?

R. All grammar corrections are accepted by the authors except for this one. Here "loosen" means the basidiomes separate from substrates, to express their relationship. We changed "loosen" to "loosening".

Reviewer 3 Report

I consider that the article has scientific merit. References are adequate and up to date. The methodology correctly describes the entire scientific procedure. The results are presented succinctly and the discussion explains and discusses the results. Therefore, I consider the article suitable for publication.

Author Response

Thank you so much for your positive comments.